# Optimal Nitrogen Supply Ameliorates the Performance of Wheat Seedlings under Osmotic Stress in Genotype-Specific Manner

**DOI:** 10.3390/plants10030493

**Published:** 2021-03-05

**Authors:** Tania Kartseva, Anelia Dobrikova, Konstantina Kocheva, Vladimir Alexandrov, Georgi Georgiev, Marián Brestič, Svetlana Misheva

**Affiliations:** 1Institute of Plant Physiology and Genetics, Bulgarian Academy of Sciences, Acad. G. Bonchev Str., bl. 21, 1113 Sofia, Bulgaria; tania_karceva@abv.bg (T.K.); aleksandrov@gbg.bg (V.A.); georgigeorgiev49@abv.bg (G.G.); s_landjeva@mail.bg (S.M.); 2Institute of Biophysics and Biomedical Engineering, Bulgarian Academy of Sciences, Acad. G. Bonchev Str., bl. 21, 1113 Sofia, Bulgaria; aneli@bio21.bas.bg; 3Department of Plant Physiology, Slovak Agricultural University, Tr. A. Hlinku 2, 949 76 Nitra, Slovakia; marian.brestic@uniag.sk

**Keywords:** chlorophyll fluorescence, dehydration, drought tolerance, nitrogen, oxidative stress, PEG, ROS, wheat genotypes

## Abstract

Strategies and coping mechanisms for stress tolerance under sub-optimal nutrition conditions could provide important guidelines for developing selection criteria in sustainable agriculture. Nitrogen (N) is one of the major nutrients limiting the growth and yield of crop plants, among which wheat is probably the most substantial to human diet worldwide. Physiological status and photosynthetic capacity of two contrasting wheat genotypes (old Slomer and modern semi-dwarf Enola) were evaluated at the seedling stage to assess how N supply affected osmotic stress tolerance and capacity of plants to survive drought periods. It was evident that higher N input in both varieties contributed to better performance under dehydration. The combination of lower N supply and water deprivation (osmotic stress induced by polyethylene glycol treatment) led to greater damage of the photosynthetic efficiency and a higher degree of oxidative stress than the individually applied stresses. The old wheat variety had better N assimilation efficiency, and it was also the one with better performance under N deficiency. However, when both N and water were deficient, the modern variety demonstrated better photosynthetic performance. It was concluded that different strategies for overcoming osmotic stress alone or in combination with low N could be attributed to differences in the genetic background. Better performance of the modern variety conceivably indicated that semi-dwarfing (*Rht*) alleles might have a beneficial effect in arid regions and N deficiency conditions.

## 1. Introduction

The effect of unfavorable environmental conditions on crop plants is significant from an economic and ecological perspective. Water availability and mineral nutrition have the greatest impact on the physiological status and productivity of agronomically important plants. Susceptibility to abiotic stress differs between species but might also be dissimilar even among the varieties of a certain crop. The harmful effects of adverse climatic conditions on plant productivity could be reduced by the selection of stress-tolerant genotypes [1]. However, this damaging impact might be decreased to some extent by improving mineral nutrition [2,3,4]. Identification of key processes which ameliorate the performance on the whole plant level is important for genotype selection and organizing breeding programs in the future [1].

Nitrogen is an essential macronutrient and a limiting factor for plant growth and productivity [5]. Suboptimal N availability could cause a significant decrease in crop yield, but excessive use of inorganic N fertilizers leads to serious environmental pollution, such as soil acidification and groundwater contamination [6].

Yet, another major restrictive factor for plant growth and productivity is water availability. Wheat is a main crop worldwide and is often exposed to water shortages which could be critical, especially during seed germination and the phase of crop establishment in early autumn [7]. Insufficient water uptake from roots on the one hand and excessive evaporation from leaves on the other lead to the development of osmotic stress. Alongside, oxidative stress appears, which induces changes in membrane structure and functioning. A common physiological parameter representative of the extent of oxidative stress is the production of reactive oxygen species (ROS), which are toxic for plants and provoke damage in DNA and proteins, lipid peroxidation, and eventually may lead to cell death [8]. An antioxidant defense system exists in plant cells which is complex and highly compartmentalized and includes enzymes and non-enzymatic components [9,10]. A common response to abiotic stress is the accumulation of proline which is supposed to sustain membrane integrity, stabilize protein structure, and reduce damage provoked by oxidative stress [11,12].

In many regions, more frequent drought episodes have been recorded during germination and crop establishment [1,13,14]. The limited nutrient availability in poor soils, in low-input production systems, as well as the restrictions on artificial N fertilizer application, pose questions about the plant capacity to manage water stress-triggered changes within plants in dependence on N supply. Given the significance of these two major resources for plants, water and N, it is essential to become more knowledge on how plants sense and respond to the combination of N and water cues. Insufficient information is available on this topic, and the reported results are inconsistent.

Photosynthesis is the main driving force for plant growth and crop grain yield. The functional state and activity of the leaf photosynthetic apparatus largely depend on the availability of N and water in plants. A high level of correlation exists between N supply and the photosynthetic efficiency in plants since N deficiency might reduce chlorophyll and other pigment content or even modify chloroplast thylakoid membranes [15,16]. Nitrogen deficiency has a negative impact on photosystem II (PSII) primary photochemistry and the electron transport rate. On the other hand, the photosynthetic efficiency decreases as relative water content and leaf water potential decrease [17]. Limitation in photosynthesis under water loss is attributed to the decrease in turgor pressure, impaired gas exchange, diminished CO_2_ assimilation, reduction in pigment content, dysfunction of the photosynthetic apparatus, enhanced metabolite fluxes, etc. [18,19]. Chlorophyll fluorescence analysis is a modern technique that serves in the identification of plant responses to abiotic stress and provides information about overall metabolic activity [20]. A contemporary study on wheat by Abid et al. [21] showed that higher N nutrition contributed to drought tolerance by maintaining higher photosynthetic activities and better preserved antioxidative defense systems during vegetative growth periods. Another recent study in wheat demonstrated that low N treatment contributed to sustained photosynthesis and plant growth under water deprivation by maintaining stomatal opening and protecting photosynthetic apparatus [22].

Conventional breeding and contemporary high-input agriculture have resulted in reduced overall genetic variation compared to the old germplasm, particularly with respect to resistance to abiotic environmental stressors [23]. On the other hand, in wheat, the introduction of della encoding mutant alleles (semi-dwarfing, or *Rht* alleles) in the tall varieties to shorten their stem increases lodging resistance and thus reduces the grain loss, which happened in the middle of the last century, and has affected the photosynthetic machinery and photosynthetic efficiency, as well as increased the plant biomass under osmotic stress [24].

In the present study, we compared two wheat genotypes—a highly productive semi-dwarf, carrier of two *Rht* alleles (Enola), and an old tall one (Slomer), a selection from a landrace, with respect to their responses to low (sub-optimal) N and drought, applied individually and in combination. The two varieties were chosen based on their contrasting reaction to N supply, with Enola displaying agronomic efficiency towards nitrogen assimilation and its allocation to the grain and Slomer exhibiting a lack of responsiveness to N supply in field conditions but having better adaptability to N deficiency [25]. Measurements of chlorophyll *a* fluorescence, water status, cell membrane stability, antioxidant, and osmoprotective capacity of leaves were used to investigate the role of N availability in the adaptation of plant metabolism and the photosynthetic apparatus to water stress. Our working hypothesis was that (1) the representatives of the old germplasm and the advanced modern breeding might have different resistance mechanisms, and (2) that proper N level would support plant growth and would lead to better performance under water shortage. Our recent findings [25] revealed that upon extreme N limitation (0.75 mM N) in the nutrient medium, the old genotype displayed better overall performance associated with better photosynthetic capacity and higher N assimilation compared to the modern variety. This study demonstrated that the consequences of limiting N nutrition were interrelated with photosynthetic metabolism. Given the importance of seedling tolerance to early-season drought for crop establishment and the limited knowledge about the N impact on plant tolerance, the objective of the present study was to compare the reactions of two wheat varieties to osmotic stress under different N supply (sub-optimal 1.5 mM and optimal 7.5 mM) evaluating biochemical parameters and photosynthetic characteristics, which could serve as stress markers and/or criteria for drought tolerance. This study will aid the better understanding of the adaptation mechanisms of wheat to environments with scarce water and/or nitrogen deficiency in poor soils and could be important for improving the drought tolerance of wheat at early developmental stages. Identifying crop strategies that develop under sub-optimal N conditions and/or water stress is an important step towards efficient agriculture.

## 2. Results

### 2.1. Growth and Physiological Parameters of Wheat Genotypes

Enola plants grown on optimal N (7.5 mM or 1 N) supply level had smaller shoot length than Slomer, but low N (1.5 mM or 0.2 N) did not have a significant impact on shoot length in either variety (Figure 1A,B). The imposition of osmotic stress caused a substantial reduction in shoot length in comparison with untreated controls, but differences between optimal and low N levels were not significant (Figure 1A,B). At optimal 1 N nutrition Enola had shorter roots than Slomer, while at reduced to 0.2 N, the reverse tendency was observed (Figure 1B). In Enola, under osmotic stress and the combination of osmotic and low N stress, root length remained as in controls with the highest values obtained under low N at optimal water supply (Figure 1C). Water and N deficiency, as well as their combined application, caused an equally significant reduction in root length in Slomer (Figure 1D).

Leaf water content did not vary between the two N nutrition levels for both varieties under water control conditions (Figure 2A,B) but decreased significantly in all plants subjected to polyethylene glycol (PEG) treatment. The greatest reduction in relative water content (RWC) was observed under the combination of low N nutrition and PEG for both genotypes. Slomer grown on sufficient N retained higher RWC than Enola after imposition of osmotic stress (Figure 2B). N supply levels, as well as osmotic stress, caused increased electrolyte leakage indicating membrane damage. The Higher Injury index was estimated under osmotic stress combined with N deficiency (Figure 2C,D). Of the two varieties, Slomer had significantly lower membrane injury than Enola, both under low N level as well as under osmotic stress.

### 2.2. Oxidative Stress and Antioxidant Capacity

The greater membrane injury in Enola compared to Slomer was accompanied by a greater amount of toxic hydrogen peroxide (H_2_O_2_) and the lipid peroxidation product malondialdehyde (MDA) in leaves of osmotically stressed plants and those grown under low N supply (Figure 3). N deficiency caused a significant increase in MDA only in Slomer while osmotic stress induced lipid peroxidation in both varieties under both N nutrition regimes (Figure 3A,B). Accumulation of H_2_O_2_ was induced by N deprivation and by osmotic stress to a similar extent in Enola, while the combination of both N and water deficiency had a greater impact compared to the singular stresses (Figure 3C). In Slomer, growth at low N induced a higher H_2_O_2_ accumulation compared to optimal N. Osmotic stress caused increased production of H_2_O_2_, with similar amounts under optimal and low N (Figure 3D).

In Enola, the total antioxidant capacity (TAC) was seriously increased by all stresses imposed. Under low N supply and the combination of the two stresses, TAC had higher values than under osmotic stress and optimal N (Figure 4A). In Slomer, osmotic stress had a greater effect on this parameter compared to low N nutrition levels. The combination of the two stresses did not increase the antioxidant capacity of Slomer compared to controls and remained substantially lower compared to the respective variant of Enola (Figure 4B). Free proline accumulated in response to both types of stresses in the leaves of plants from the two wheat varieties (Figure 4C,D). Osmotic stress induced a greater increase in proline levels (more pronounced for Enola) compared to N deficiency. However, the highest proline content was found in both genotypes/varieties experiencing N and water shortage simultaneously.

### 2.3. Photosynthetic Pigments

Regarding the total leaf chlorophyll content in control plants, Slomer had higher values than Enola independently of N supply (Figure 5A,B). Further, N and water deficiency caused a decrease in these pigments but to a smaller extent in Slomer than in Enola. Carotenoid content in Slomer was not affected by low N-only treatment, while it was reduced in Enola (Figure 5C,D). Osmotic stress caused a stronger reduction in carotenoids than N deficiency in both cultivars, while the combination of the two stresses had a stronger impact on carotenoid content in Slomer.

### 2.4. Photosynthetic Performance

For the purposes of the present investigation, several parameters calculated from chlorophyll *a* fluorescence were investigated. The most commonly used parameter in the fluorescence analysis is maximal quantum yield of PSII photochemistry (F_v_/F_m_), representing the maximum efficiency with which light absorbed by the light-harvesting complexes of PSII is converted into photochemical processes. This parameter can be measured only when the leaf is in a fully relaxed, dark-adapted state. Under N deficiency, Enola and Slomer did not show significant differences in F_v_/F_m_ values (Figure 6A,B). There were no significant differences between the values for control plants and those under osmotic stress at optimal N. When plants were simultaneously subjected to N and water deficiency, the values of F_v_/F_m_ demonstrated significant differences between cultivars, with Enola maintaining higher values. Enola had slightly higher values of F_v_/F_0_ than Slomer under drought combined with N deficiency (Figure 6C,D).

The parameter Ф_PSII_ shows the efficient quantum yield of PSII photochemistry [26,27]. Control values of this parameter were similar in both varieties (Figure 7A,B). Under N scarcity, a small decrease was evidenced in the two varieties, and again values did not differ significantly regarding this parameter between genotypes. Under osmotic stress at optimal N, Ф_PSII_ decreased more in Enola compared to Slomer, but when both stresses were applied simultaneously, Enola displayed better PSII photochemistry than Slomer (Figure 7A,B). Photochemical quenching q_P_ was used to quantify the fraction of the open PSII reaction centers with oxidized primary quinone electron acceptor QA [28]. Slomer had slightly higher q_P_ values than Enola under the combination of N deficiency and dehydration (Figure 7C,D). On the other hand, the values of this parameter were similar in both varieties when plants were subjected to two N supply levels (Figure 7C,D). The parameter Φ_NPQ_ represents the quantum yield of regulated non-photochemical energy loss in PSII, i.e., thermal energy dissipation by downregulation in PSII [27]. Its levels increased significantly upon N and water deficiency only in Enola (Figure 8A,B). Therefore, under the combination of the two stresses, Enola exhibited much higher Φ_NPQ_ than Slomer.

Φ_NO_ represents the quantum yield of non-regulated energy loss in PSII, a loss process due to PSII inactivity [27]. This parameter increased more strongly in Slomer than in Enola under the combination of both stresses (Figure 8C,D). In addition, the increase in two parameters: Φ_NPQ_ and Φ_NO_ (Figure 8), was associated with a decrease in the quantum efficiency of photochemical energy conversion in PSII (Figure 8A,B). According to the results of chlorophyll fluorescence analysis of the two wheat genotypes in our experiments, the quantum efficiency of PSII (Φ_PSII_) was better in Enola than Slomer when plants were exposed to the combination of both stresses—N deficiency and dehydration (Figure 7A,B).

Alterations in the photooxidation of P700 to P700^+^, corresponding to PSI photochemical activity [29,30], were determined on dark-adapted wheat leaves by following the far-red light-induced absorbance changes at 830 nm (ΔA_830_). Results showed that the control values of this parameter were similar in Enola and Slomer regardless of N supply (Figure 9). Furthermore, the optimal N supply (1 N) in the nutrient media positively influenced the relative oxidation of P700 to P700^+^ (ΔA_830_) under osmotic stress in both varieties (Figure 9). In addition, when plants were subjected simultaneously to osmotic stress and N deficiency, the amount of P700^+^ decreased, but it maintained higher values in Slomer than in Enola (Figure 9).

## 3. Discussion

N availability is the most significant factor influencing crop yield. Besides its important role in growth and developmental processes, N is also involved in plant responses toward stressful environmental conditions [31]. Since great quantities of N take part in building the components of the photosynthetic apparatus, adequate N supply affects the proper functioning of photosynthesis. On the other hand, N assimilation requires ATP, reducing equivalents and C skeletons derived from primary photochemical reactions and CO_2_ fixation, respectively. Thus, plant growth and biomass production are dependent upon the fine interplay between C and N metabolism [31].

Genotypes with high N use efficiency that can grow well on soils with low N availability are of particular importance to sustainable agriculture to decrease the use of fertilizers which convey negative consequences to agroecosystems [32]. Water availability is yet another prerequisite for optimal crop growth, development, and yield. Contemporary high-yielding semi-dwarf wheat varieties demand greater amounts of water and N-fertilizer, and they are sensitive to both water and N deficiency. Usually, in the field, abiotic factors act in combination [33]. Our understanding of these processes has remained fragmentary because the comparison of genotypes has rarely been carried out simultaneously at agronomic and physiological levels [2]. Searching for some safe ways for the environment to alleviate the negative effects of dehydration on crop growth and development has become imperative in recent years. Under decreased N supply, the old variety Slomer showed lower susceptibility to oxidative stress, which could probably be attributed to its broader intrinsic genetic variation for resistance to abiotic and biotic stresses [23]. Тhe imposed osmotic stress induced stronger oxidative damage and forced defense mechanisms in response to a higher level of dehydration (more reduced leaf RWC) more distinctly in Enola compared to Slomer (Figure 2).

In both wheat varieties, N deficiency increased lipid peroxidation and H_2_O_2_ accumulation, but water deficiency additionally aggravated oxidative damage and membrane injury, in particular, as these oxidative stress markers increased more strongly in Enola than in Slomer (Figure 3). It has been reported recently that higher N availability enhanced N metabolism and contributed to improving stress tolerance of the photosynthetic apparatus by preventing cell membrane damage and supporting more effective energy dissipation and ROS scavenging in rice [34] and winter wheat [33].

Plant antioxidant capacity is greatly dependent upon N availability. Most probably, higher N improves stress tolerance of plants via the enhancement of the antioxidant ability and inhibition of lipid peroxidation. On the contrary, low N attenuates ROS scavenging systems and increases oxidative stress in leaves, and this was clearly evidenced in Slomer [35,36]. Low N supply induced higher lipid peroxidation and H_2_O_2_ content compared to optimal N in both varieties, and water scarcity aggravated oxidative membrane injury at low N, more obvious in Enola than in Slomer (Figure 3). The greater membrane injury in Enola compared to Slomer was accompanied by a greater amount of toxic H_2_O_2_ and lipid peroxidation (Figure 2 and Figure 3). In Enola, the total antioxidant capacity was strongly increased under both stresses imposed in comparison to Slomer (Figure 4).

Although the photosynthetic performance of Slomer was better under drought stress and optimal N supply, when N deficiency was combined with osmotic stress, the modern variety Enola appeared to be more adaptive under the limiting conditions (Figure 6 and Figure 7).

Relatively higher chlorophyll content in the old variety Slomer under both stresses did not provide higher photosynthetic efficiency (Figure 5, Figure 6 and Figure 7). On the contrary, reduced chlorophyll content in the modern variety Enola probably represented an effective strategy to prevent photosynthesis from photoinhibition and oxidative damage under stress. It helped plants avoid excessive light energy absorption and improve the quantum efficiency of PSII, thus determining a higher photosynthetic rate.

Our results revealed that the combination of the two stresses (low N and water deficiency) led to a more significant decrease in the parameters of maximal and potential photochemical activity of PSII, the ratios F_v_/F_m_ and F_v_/F_0_, respectively (Figure 6) than any of the stresses imposed individually. The strongest decrease in the effective quantum yield of PSII photochemistry (Φ_PSII_) was also observed under the combination of water and N deficiency, more obvious in the older variety Slomer than in the modern Enola (Figure 7). It could be speculated that a better N supply improved the photosynthetic performance under dehydration in both wheat cultivars but more pronounced in Slomer. The reduction in Φ_PSII_ indicated a decline in excitation energy captured by reaction centers and decreased electron transport activities in PSII. Drought-induced inhibition of the effective quantum yield of PSII photochemistry (Φ_PSII_) indicated that the electron-transport processes were downregulated in both wheat cultivars depending on N supply as Enola was more resistant than Slomer under low N supply (Figure 7). In accordance with our data, a previous study [26] also demonstrated that drought stress strongly suppressed the utilization of the excitation energy by photochemistry, causing an inhibition of the electron transport rate and Ф_PSII_, as well as reduction of the PSII acceptor side (q_p_), connected with an increased thermal dissipated energy (i.e., higher non-photochemical quenching, NPQ values). More recent studies have also shown that drought stress had a significant negative effect on photosynthetic characteristics and chlorophyll content in rice plants [34,37].

Furthermore, non-photochemical quenching has been proposed to be an important mechanism to diminish the ROS production in the photosynthetic membranes and plays a major role in the protection of PSII from photodamage [38]. The quantum yields of pH-dependent non-photochemical quenching (Ф_NPQ_) were found to be associated with the acidification of the thylakoid lumen and regulation of light-harvesting processes resulting in activation of violaxanthin de-epoxidase [28,39]. Under control conditions, the quantum yield of regulated non-photochemical energy loss in PSII (Ф_NPQ_) was higher in Enola than Slomer at normal N supply (Figure 8A,B), as observed previously [25]. However, under low N (0.2 N), both cultivars showed different alterations than those observed under extremely low N (0.1 N) [25]. Moreover, an increased Φ_NPQ_ upon low N and drought stress observed for Enola denoted higher regulated thermal dissipation of excitation energy, which could be regarded as a mechanism for better protection of the PSII photochemistry against oxidative damage (Figure 8A,B). Similar results were obtained in winter wheat and rice [33,34]. Moreover, the acidification of the thylakoidal lumen has been proposed to be a defense mechanism of the photosynthetic apparatus under drought stress causing the activation of non-photochemical processes [29]. On the other hand, the quantum yields of non-regulated energy loss in PSII (Ф_NO_) were more enhanced in Slomer under the combination of both stresses (Figure 8C,D).

In addition to the PSII photoinhibition, a lower capacity for P700 photooxidation has been observed only under low N levels (0.2 N) and osmotic stress for both cultivars (Figure 9). Previous studies have also demonstrated that the drought stress causes the oxidation of the reaction center of the chlorophyll of PSI (P700), concomitant with a decrease in the quantum efficiencies of PSII accompanied by increases in NPQ to protect P700 from over-reduction in wheat and rice [40,41]. These results suggested that the drought stress responses of the photosynthetic electron-transport reactions are closely associated with the oxidization of P700. Data presented here showed that the extent of P700 oxidation was stimulated at optimal N supply and dehydration (Figure 9). Since P700 oxidation is thought to suppress reactive ROS production and so protects PSI from photoinhibition [41], the drought-induced responses of the PSI photooxidation under optimal N conditions observed here could be a defense mechanism to protect PSI reaction centers via the oxidation of P700. However, under the combination of both stresses, the amount of P700^+^ decreased more pronounced for Enola than for Slomer (Figure 9).

All the results presented here suggest that the impact of osmotic stress on wheat photosynthetic apparatus was most deleterious under lower N levels. This is expected since nutrient deficiency alone has been shown to inactivate the PSII reaction centers, inhibit photosynthetic function, and enhance thermal energy dissipation [33,34]. In contrast, optimal N supply likely contributes to plants’ ability to cope with drought stress. In addition, it is proposed that the improved photosynthetic characteristics are associated with the increase in crop yields [14].

It has been established that upon desiccation, the photon flux density is higher while the demands for ATP and reduction equivalents for assimilatory processes are decreased. Thus, the excessive light energy could lead to the formation of potentially harmful ROS, which should be eliminated through specially developed defense mechanisms [9]. The antioxidant capacity of plants comprises enzymatic reactions and biosynthesis of non-enzymatic low molecular metabolites, such as ascorbate, reduced glutathione, carotenoids, flavonoids, and proline [10,42]. These antioxidant compounds contribute to stress relief by acting as radical scavengers that aid ROS detoxification and the protection of subcellular structures. The stress-induced imbalance between light energy absorption and its consumption subsequently stimulates photooxidative damage of photosynthesis, as well as peroxidation of membrane lipids [10,34]. Therefore, maintaining an appropriate level of ROS and preventing impairment of the photosynthetic apparatus is essential for the proper functioning of photosynthesis.

The potential role of N is not only related to maintaining the photosynthetic machinery by providing the building blocks of proteins and chlorophyll but N compounds and nitrate (NO_3_^−^), in particular, participate in the regulation of stomatal opening. Since N assimilation consumes ATP and reducing power (NAD(P)H), it can partly dissipate excessive captured light energy and mitigate photoinhibition induced by water stress through the synthesis of nitrogenous compounds, e.g., amino acids and soluble proteins, and also by promoting the accumulation of osmotic compounds, such as proline [12]. Thus, proper functioning of N metabolism could contribute to improving photosynthesis acclimation to water stress.

Proper N levels support regular plant growth and help plants defend against stress, possibly by mitigating the adverse effects on photosynthesis. In Enola, the increased resistance of photosynthesis to water stress at low N in comparison to Slomer seemed to result from enhanced functioning of the antioxidant defense systems (Figure 4). Since N assimilation acts as an important alternative sink of electrons and excessive excited energy, it probably minimizes photoinhibition and photodamage of photosynthesis. This is beneficial in combination with stimulated CO_2_ fixation under stomatal limitation, which accompanies strong dehydration [43,44]. The study of [37] on rice plants also suggested that N supply appeared to promote photosynthetic tolerance to water stress via affecting CO_2_ diffusion, antioxidant capacity, and osmotic adjustment since a major part of the assimilated N was invested in the photosynthetic machinery. The significant effect of N and water stress interactions on photosynthetic activity implies that N is essential for regulating the adaptation of photosynthetic apparatus to dehydration.

We have hypothesized that plants supplied with optimal N would have better growth and photosynthetic performance than those supplied with lower N, and it would give an advantage under water limiting conditions. Our results also indicated that different varietal responses to N supply would contribute to different performances under drought stress. Sufficient N supply facilitates the assimilation of stored nitrates under water deficiency, which could partly contribute to mitigating photoinhibition of photosynthesis commonly induced by dehydration. Thus, proper N nutrition could clearly be considered an effective tool for osmotic stress management. It was also documented recently that photosynthetic mechanisms against stress might show some natural variation, especially regarding NPQ and protection against PSII photodamage [44].

Effects of the genetic background should also be considered. The elite semi-dwarf variety Enola carries mutant alleles of two *Rht* genes—*Rht-B1b/d* and *Rht8* [45], while in Slomer, the old variety of tall stature, wild alleles are present at both *Rht* loci. The contribution of *Rht-B1* genes to enhanced tolerance to water shortage in wheat was recently suggested to be due to modulated plant antioxidant system, enhanced osmoregulation, better-sustained membrane integrity, more effective performance of the photosynthetic machinery [46]. Moreover, the role of *Rht-**B1* genes was proposed in the protection of the photosynthetic apparatus under cadmium [47] and salinity [48] stress and better N supply had a beneficial effect on the alleviation under cadmium stress [49]. Recently, it was reported that under osmotic stress, other *Rht* mutants (*Rht13*) were able to maintain their water balance more effectively compared to the wild type counterparts (*rht* lines) due to the significantly lower transpiration rates, lower rates of leaf water loss, and improved water uptake ability of roots [24]. The *Rht13* seedlings also displayed relatively higher photosynthetic rates and higher reduction in the stomatal conductance under osmotic stress than the *rht* lines. Our present results support the possibility for the beneficial pleiotropic effect of the Rht genes on plant tolerance to osmotic stress. Under the combination of N deficiency and dehydration, Enola had better physiological performance compared to Slomer, which could be an indication of the beneficial pleiotropic effect of the *Rht* genes on abiotic stress resistance previously discussed by some authors [24,46,50].

## 4. Materials and Methods

### 4.1. Plant Material and Growing Conditions and Treatments

Two Bulgarian bread wheat varieties, Slomer and Enola, with contrasting reaction to N supply, were used. Enola is a modern semi-dwarf variety carrying a combination of two height-reducing genes, *Rht8 + Rht − B1b/d* [35]; highly productive, displaying agronomic efficiency towards nitrogen assimilation and its allocation to the grain [35]. Slomer is an old tall variety with wild-type alleles at *Rht* loci, which exhibited a lack of responsiveness to nitrogen supply in field conditions [45] but had better adaptability to N deficiency. Seeds were superficially sterilized with 5% Na hypochloride for 10 min, thoroughly washed with running tap water, and germinated in Petri dishes on moist filter paper at 21 °С in darkness. Three-day-old seedlings were transferred to the nutrient solution. For optimal (referred to as 1 N) nitrogen conditions, the nutrient solution contained the following macronutrients: 2.5 mM Ca(NO_3_)_2_·4H_2_O, 2.5 mM KNO_3_, 2 mM MgSO_4_·7H_2_O, 1 mM KH_2_PO_4_, or a total of 7.5 mM nitrates. For nitrogen deficiency conditions, N levels were reduced to 1/5th of controls and referred to as 0.2 N or low N: 0.5 mM Ca(NO_3_)_2_·4H_2_O and 0.5 mM KNO_3_ giving an overall of 1.5 mM of nitrates, while all other nutrient components remained as in controls. Micronutrient composition was equal for both variants: 9 µM Fe-EDTA, 4.0 µM H_3_BO_4_, 1.0 µM CuSO_4_, 0.9 µM ZnSO_4_, 1.8 µM MnCl_2_, 0.2 µM NaMoO_4_. Plants were grown in a climatic chamber with 22/18 °C day/night temperature, respectively, 12 h photoperiod, irradiance of 250 μmol m^−2^ s^−1^ and 70% relative humidity for 10 days. At the 11th day part of the plants were subjected to 72 h of osmotic stress with 25% PEG 6000 (Sigma–Aldrich), producing a water potential of −1.0 MPa. Control plants remained on the nutrient solution (1 N and 0.2 N, respectively, for the two variants).

### 4.2. Growth Parameters and Determination of Relative Water Content

Root and shoot length of ten plants from each variant and genotype were measured. Relative water content (RWC) was measured according to Turner [51] and estimated as:RWC = (FW − DW)/(TW − DW) × 100(1)
where FW is the leaf fresh weight, TW is the weight at full turgescence after floating leaves in water for 24 h at room temperature in darkness, and DW is the dry weight after drying at 80 °С to a constant weight.

### 4.3. Electrolyte Leakage Measurement

Leaves were cut to pieces of 2 cm length, excluding tip and base, washed with distilled water to eliminate injured tissues, and 15 pieces were incubated in 15 mL distilled water for 24 h at room temperature. The conductivity of the solutions was measured once and then once again after boiling the samples for 30 min followed by cooling at room temperature.

The injury index was assessed by the formula:I = 1 − (1 − T_1_/T_2_)/(1 − C_1_/C_2_) × 100(2)
where T_1_ and T_2_ are the values measured before and after boiling of leaves of PEG-treated and plants grown with low N, and C_1_ and C_2_ are the corresponding values of control plants [52].

### 4.4. Biochemical Analysis

All measurements were performed on leaves of control and PEG-treated wheat plants. Leaf samples of 0.5 g fresh weight were collected, and free proline was evaluated by the ninhydrin-acetic acid method of [53]. Malondialdehyde (MDA) content was determined as thiobarbituric acid-reagent product according to [54], and the concentration was calculated using the extinction coefficient of 155 mM^−1^cm^−1^. Hydrogen peroxide was estimated spectrophotometrically according to [55]. The amount of hydrogen peroxide was calculated from a standard curve prepared with known concentrations of H_2_O_2_. Total antioxidant capacity (TAC) was estimated by the ferric reducing antioxidant power (FRAP) assay described in [56]. The method is based on the reduction of a ferric-tripyridyl triazine complex to its ferrous-colored form in the presence of antioxidants. Freshly prepared FRAP reagent containing 10 mM TPTZ (2,4,6-tripyridyl-s-triazine) solution in 40 mM HCl, 20 mM FeCl_3_ and 0.3 M acetate buffer, (pH 3.6) was warmed at 37 °C. Aliquots of 100 μL sample were mixed with 3 mL FRAP reagent, and the absorbance of the reaction mixture was measured spectrophotometrically at 593 nm after 10 min incubation at 37 °C. A calibration curve with five concentrations of FeSO_4_·7H_2_O (1000, 750, 500, 250, 125 μmol/L) was prepared. Values were expressed as the concentration of antioxidants having a ferric reducing ability equivalent to that of 1 µmol/L FeSO_4_.

### 4.5. Pigment Analysis

Fresh leaf samples were homogenized in ice-cold 80% (*v*/*v*) acetone and were then centrifuged at 5000× *g* for 5 min at 4 °C. The supernatant was measured spectrophotometrically using Specord 210 plus (Analytic-Jena AG, Jena, Germany), recording the absorptions at 470, 646.8, and 663.2 nm. The total chlorophyll content (Chl *a* and Chl *b*) and total carotenoids (Car) were calculated by the equations of [57].

### 4.6. Chlorophyll Fluorescence Measurements

In vivo chlorophyll fluorescence at room temperature was measured in dark-adapted leaves using a PAM fluorometer (model 101/103, Waltz GmbH, Effeltrich, Germany). The minimum fluorescence level (F_0_) was detected after at least 20 min of dark adaptation by applying very weak modulated light (0.02 µmol m^−2^ s^−1^ photosynthetic photon flux density, PPFD). The maximum fluorescence levels in dark-adapted state (F_m_), light-adapted state (F_m_′), and after dark relaxation following the light period (F_m_″) were obtained by illuminating the leaf samples with a saturating short flash (2000 µmol m^−2^ s^−1^ PPFD, 0.8 s) and steady-state chlorophyll fluorescence (F_s_) was recorded. Photosynthesis was induced by continuous illumination of leaves with actinic light at 250 µmol m^−2^ s^−1^ PPFD, which corresponds to the illumination during plant growth. The steady-state fluorescence level (F′) was estimated 7–8 min after turning on the actinic light. The following parameters were determined according to [27]: maximum quantum efficiency of PSII in dark-adapted state (F_v_/F_m_ = (F_m_ − F_0_)/F_m_), the ratio of photochemical to non-photochemical processes in PSII or potential photosynthetic activity (F_v_/F_0_) [26], effective quantum yield of PSII photochemistry (Φ_PSII_ = (F_m_′ − F′)/ F_m_′), photochemical quenching coefficients (q_P_ = (F_m_′ − F′)/ F_v_′), the quantum yield of regulated non-photochemical quenching (Ф_NPQ_ = F_s_/F_m_′ − F_s_/F_m_), and the quantum yield of non-regulated energy dissipated in PSII (Ф_NO_ = F_s_/F_m_).

### 4.7. P700 Redox State Measurements

Steady-state P700 photooxidation (P700^+^) was measured by the illumination of dark-adapted detached leaves with a far-red (FR) light supplied by a photodiode (102-FR Waltz, Effeltrich, Germany). Redox changes in P700 were monitored as FR light-induced absorbance changes at approximately 830 nm (ΔA830) measured with a PAM 101/103 fluorometer equipped with an ED-800T emitter detector, as described in [49].

### 4.8. Statistical Analysis

Two independent experiments were performed, and each parameter was assessed in at least three technical replications (*n* is indicated in figure captions). Data are presented as means ± SE. The significance of differences between genotypes and treatments was assessed by ANOVA followed by Tukey’s test for each parameter. Values with the same letter were not significantly different at *p* < 0.05.

## 5. Conclusions

Overcoming the negative effects of dehydration on the performance of economically important crops has become imperative for sustainable agriculture. Damages on plant productivity could be reduced by the selection of stress-tolerant genotypes. Data in the current study showed for the first time that a modern wheat variety (Enola) retained better PSII photochemistry under combined low N and osmotic stress conditions than an old variety (Slomer), despite higher oxidative damage and membrane injury observed in the leaves of the former. This could be due to the increased proline synthesis, antioxidant activity, as well as higher values of Ф_NPQ_ which most probably participate in protective mechanisms for coping with abiotic stresses in the modern wheat variety. Therefore, both wheat genotypes showed different tolerance strategies against the applied osmotic stress alone or in combination with N deprivation, possibly owing to differences in the genetic background. Higher quantum yields of PSII photochemistry and regulated non-photochemical quenching under osmotic stress and low N supply evidenced in the semi-dwarf variety Enola indicated that *Rht* alleles may have beneficial roles in arid regions and N deficiency conditions, but further detailed studies using specific genetic resources, such as *Rht* near isogenic lines, coupled with experiments in the field will be required to fully determine their effects.

## Figures and Tables

**Figure 1 plants-10-00493-f001:**
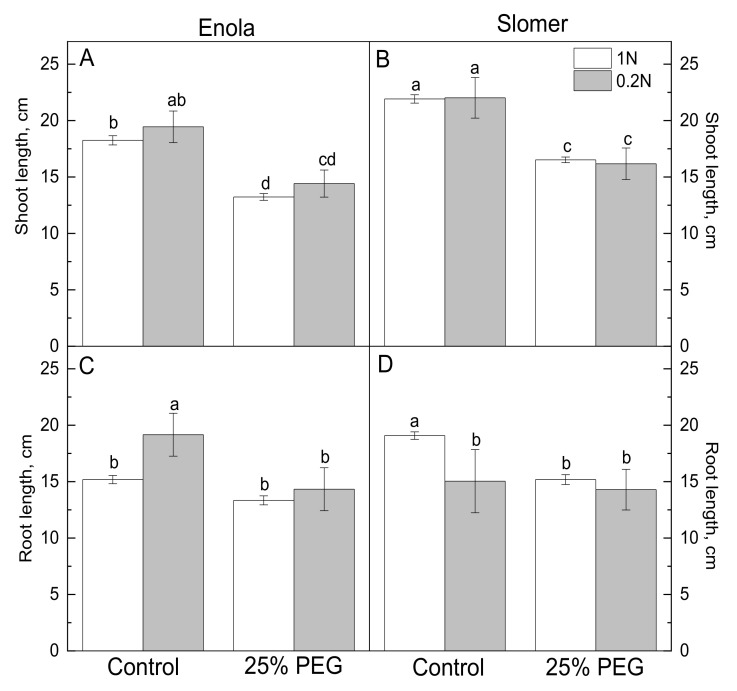
Growth parameters assessed by shoot (**A**,**B**) and root (**C**,**D**) length of two wheat varieties (Enola and Slomer) supplied with optimal (1 N) or low (0.2 N) nitrogen in the medium and subjected to osmotic stress with 25% PEG. Different letters represent significant differences at *p* < 0.05 (*n* = 10).

**Figure 2 plants-10-00493-f002:**
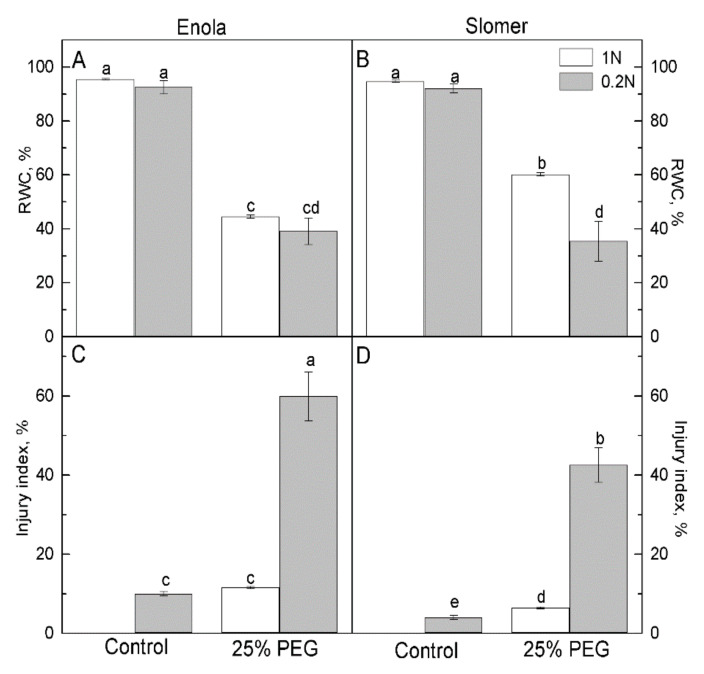
Changes in relative water content (RWC) (**A**,**B**) and membrane injury index (**C**,**D**) in leaves of two wheat varieties (Enola and Slomer) grown with optimal (1 N) and reduced (0.2 N) nitrogen supply and subjected to osmotic stress with 25% PEG. Different letters represent significant differences at *p* < 0.05 (*n* = 6).

**Figure 3 plants-10-00493-f003:**
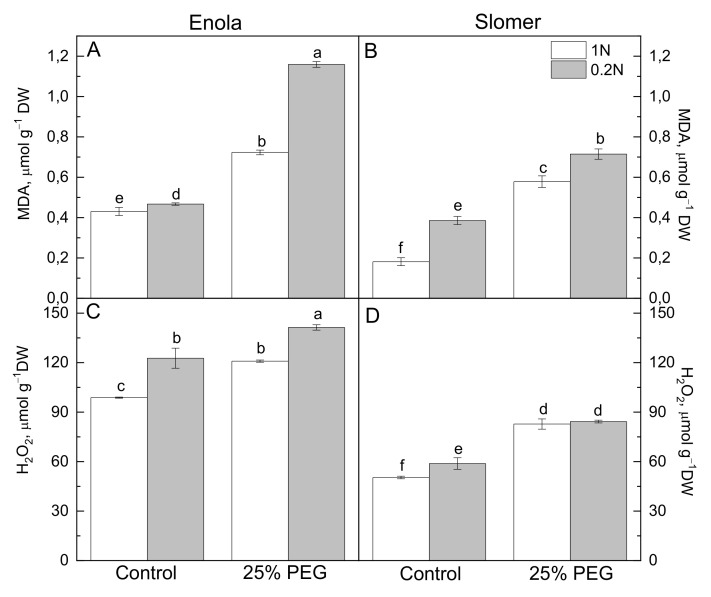
Amounts of MDA (**A**,**B**) and H_2_O_2_ (**C**,**D**) in leaves of two wheat varieties (Enola and Slomer) grown with optimal (1 N) and reduced (0.2 N) nitrogen supply and subjected to osmotic stress with 25% PEG. Different letters represent significant differences at *p* < 0.05 (*n* = 8).

**Figure 4 plants-10-00493-f004:**
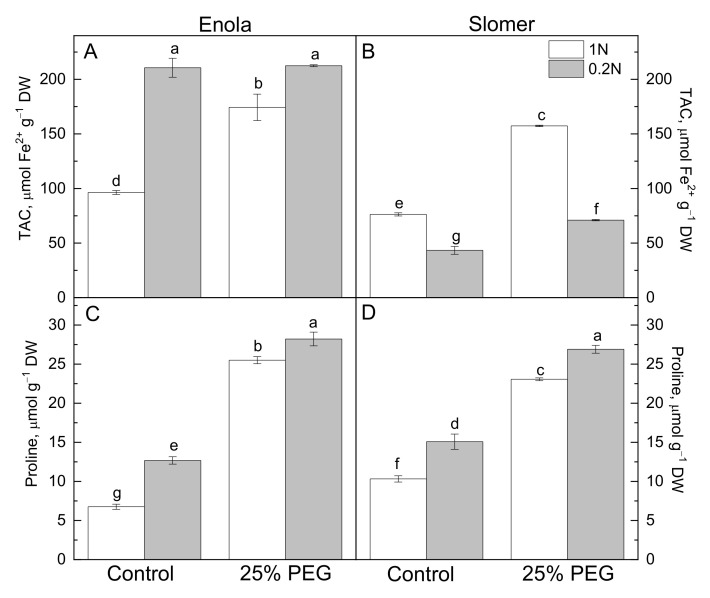
Total antioxidant capacity (TAC) (**A**,**B**) and proline content (**C**,**D**) in leaves of two wheat varieties (Enola and Slomer) grown with optimal (1 N) and reduced (0.2 N) nitrogen supply. Different letters represent significant differences at *p* < 0.05 (*n* = 8).

**Figure 5 plants-10-00493-f005:**
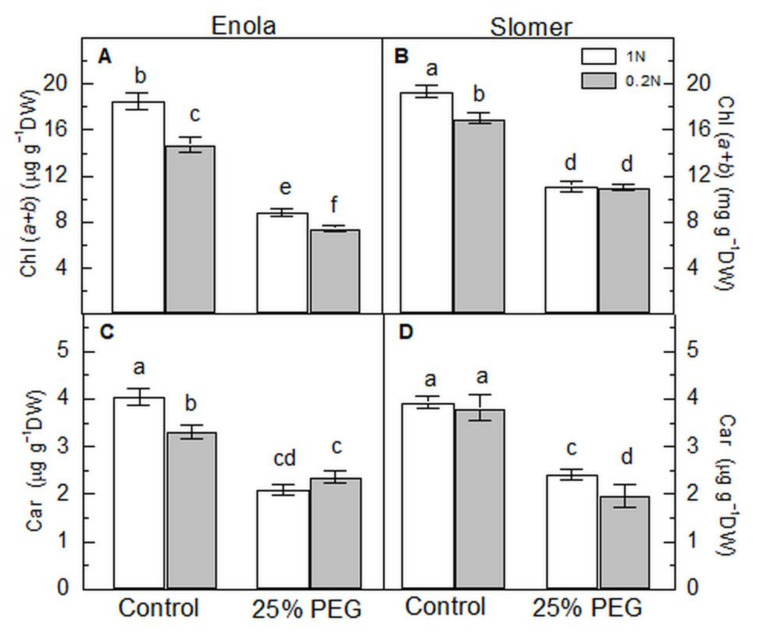
Leaf pigment content: total chlorophyll (Chl *a + b*) (**A**,**B**) and carotenoids (Car) (**C**,**D**) of two wheat varieties (Enola and Slomer) grown with optimal (1 N) and reduced (0.2 N) nitrogen supply and subjected to osmotic stress with 25% PEG. Different letters represent significant differences at *p* < 0.05 (*n* = 8).

**Figure 6 plants-10-00493-f006:**
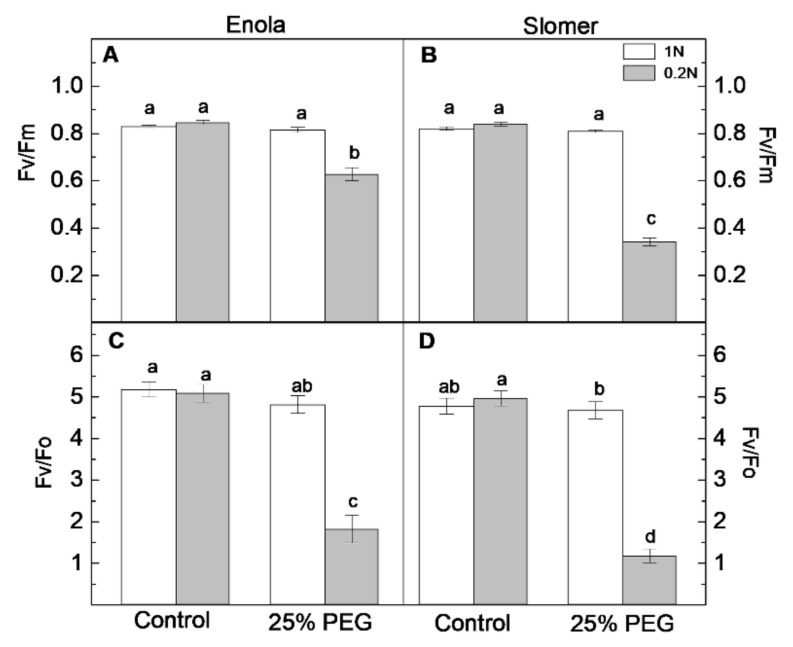
Chlorophyll fluorescence parameters: (**A**,**B**) maximum quantum efficiency of photosystem II (PSII) (F_v_/F_m_) and (**C**,**D**) potential photosynthetic activity (F_v_/F_0_), measured in leaves of two wheat cultivars (Enola and Slomer) grown on optimal (1 N) and reduced (0.2 N) nitrogen supply and subjected to osmotic stress with 25% PEG. Different letters represent significant differences at *p* < 0.05 (*n* = 8).

**Figure 7 plants-10-00493-f007:**
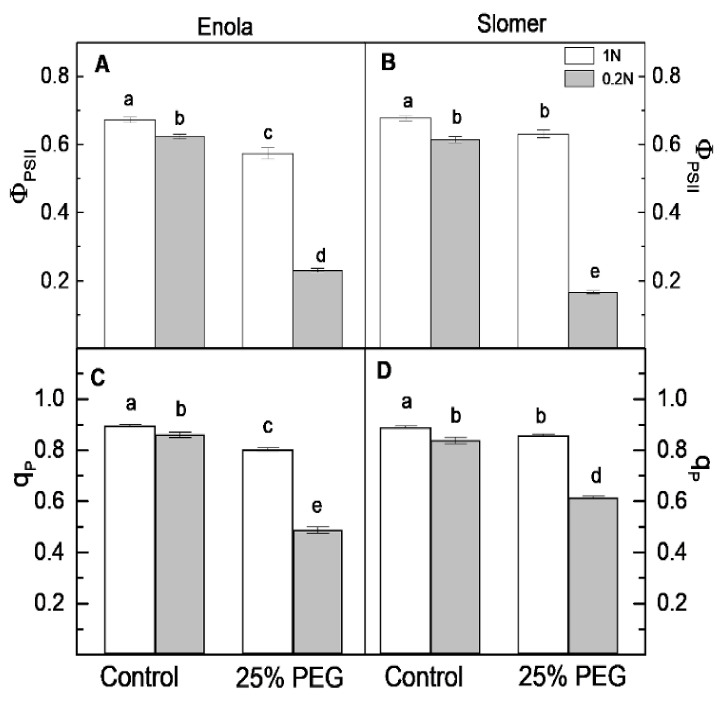
Effective quantum yield of PSII photochemistry, Φ_PSII_ (**A**,**B**) and photochemical quenching coefficient, q_P_, (**C**,**D**) estimated in leaves of two wheat varieties (Enola and Slomer) grown with two nitrogen supply levels (1 N and 0.2 N) and subjected to osmotic stress with 25% PEG. Different letters represent significant differences at *p* < 0.05 (*n* = 8).

**Figure 8 plants-10-00493-f008:**
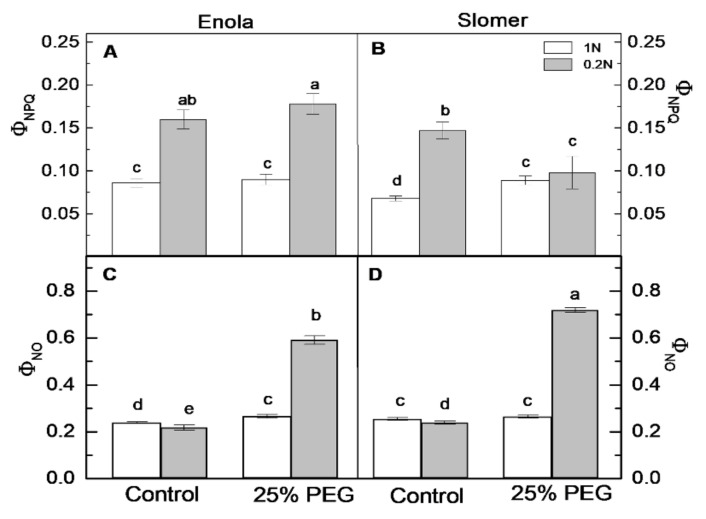
Non-photochemical quenching of chlorophyll fluorescence, Φ_NPQ_ (**A**,**B**) and quantum efficiency of non-photochemical quenching, Φ_NO_ (**C**,**D**) in leaves of two wheat varieties (Enola and Slomer) grown with two nitrogen supply levels (1 N and 0.2 N) and subjected to osmotic stress with 25% PEG. Different letters represent significant differences at *p* < 0.05 (*n* = 8).

**Figure 9 plants-10-00493-f009:**
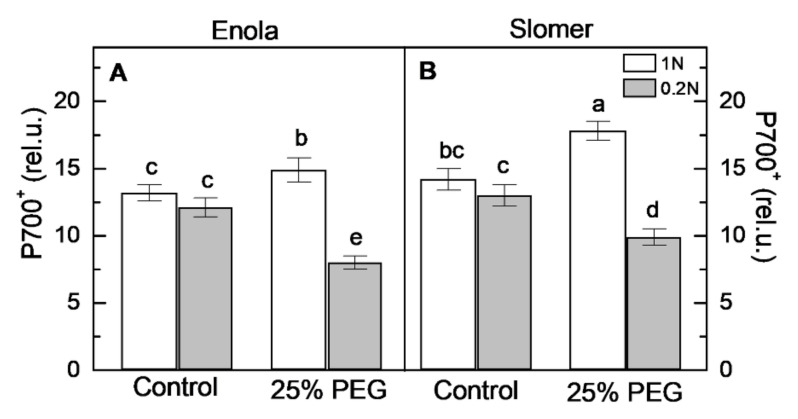
Photooxidation of P700 (P700^+^) measuring as ΔA_830_ in leaves of two wheat varieties (Enola and Slomer) grown with two nitrogen supply levels (1 N and 0.2 N) and subjected to osmotic stress with 25% PEG. Different letters represent significant differences at *p* < 0.05 (*n* = 8).

## Data Availability

Not applicable.

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
