# Peer review of "Optimal Nitrogen Supply Ameliorates the Performance of Wheat Seedlings under Osmotic Stress in Genotype-Specific Manner"

_plants, 2021, doi:10.3390/plants10030493_

Round 1

Reviewer 1 Report

Review

 Kartseva et al: Optimal Nitrogen Supply Ameliorates the Performance of Wheat Seedlings under Osmotic Stress in Genotype-Specific Manner

This study describes the effects of low-N and osmotic stress alone and in combination in two wheat cultivars, an old landrace variety and a new semi dwarf modern selection .It was concluded that the two varieties follow different strategies in their stress responses. One of the most significant differences was found in the better photosynthetic performance of the modern variety under high N-supply, and among other responses, the differences were attributed to the genetic background i.e. the presence of the semi-dwarfing (Rht) alleles. Although the modern variety was more sensitive at membrane level and to oxidative stress, they were compensated by a more efficient antioxidative system, compared to the landrace. However, under N-limitation, the old, tall landrace gave better performance.

These findings are the strength of the paper.

In general, the paper is a traditional study, the approach of the problem is correct, the methodology used is appropriate.

The results are focusing on the growth and water parameters (Figs. 1 and 2,) on the oxidative stress (Figs 3 and 4) and mainly on the photosynthesis (Figs 5-9) and so, at the first glance, the reader is left uninformed about the N-metabolism. However, after re-reading the introduction, it turns out that this part of the study based on the self-reference (25),  Kocheva et al.: Nitrogen assimilation and photosynthetic capacity of wheat genotypes under optimal and deficient nitrogen supply. Physiol. Mol. Biol. Plants 2020, 26(11), 2139–2149. DOI: 611 10.1007/s12298-020-00901-3.

Therefore, I suggest to summarize shortly the differences in N-metabolism (nitrate reductase activity and further steps) between the two varieties (more explicitly after the citation /25/ in Introduction.

Further remarks:

Materials and Methods

Regarding osmotic stress: what is the osmotic potential (mOsm) and/or water potential (MPa) of the used 25% PEG 6000 solution?

Results

Figures and Figure legends are in order.

Fig. 1.: Did you observe symptoms of N-deficiency in 14-d-old 0.2N plants?           

  • (Until 8-9 days plants are not fully autotrophic, they rely on stored sources in the caryopsis, therefore one cannot expect large effects of the nutrient solution on growth.)

Author Response

The results are focusing on the growth and water parameters (Figs. 1 and 2,) on the oxidative stress (Figs 3 and 4) and mainly on the photosynthesis (Figs 5-9) and so, at the first glance, the reader is left uninformed about the N-metabolism. However, after re-reading the introduction, it turns out that this part of the study based on the self-reference (25), Kocheva et al.: Nitrogen assimilation and photosynthetic capacity of wheat genotypes under optimal and deficient nitrogen supply. Physiol. Mol. Biol. Plants 2020, 26(11), 2139–2149. DOI: 611 10.1007/s12298-020-00901-3.

A: We are very grateful to Reviewer 1 for thorough reading and valuable comments, and we have tried to fulfil all requirements as follows:

Q: Therefore, I suggest to summarize shortly the differences in N-metabolism (nitrate reductase activity and further steps) between the two varieties (more explicitly after the citation /25/ in Introduction.

A: In Introduction (page 3, line 103-131) we have briefly summarized the differences between the two genotypes which were described in our previous work as the Reviewer properly suggested. The inserted text is marked with Track changes.

Q: Further remarks:

Materials and Methods

Regarding osmotic stress: what is the osmotic potential (mOsm) and/or water potential (MPa) of the used 25% PEG 6000 solution?

A: In Materials and Methods (page 14, line 477-478), we have added the water potential of the 25% PEG solution as the Reviewer appropriately suggested.

Q: Results

Figures and Figure legends are in order.

 Fig. 1.: Did you observe symptoms of N-deficiency in 14-d-old 0.2N plants? (Until 8-9 days plants are not fully autotrophic, they rely on stored sources in the caryopsis, therefore one cannot expect large effects of the nutrient solution on growth.)

A: Indeed, we did not observe reduced shoot growth due to low N supply. However, our results showed that low (0.2N or 1.5 mM) N supply had apparent impact on photosynthetic pigments and early stress markers such as proline and hydrogen peroxide accumulation of both genotypes. Previous study on wheat also evidenced a close relationship between chlorophyll and nitrogen content in leaves, because N is a structural element of chlorophyll and protein molecules (Bojović and Marković, Correlation between nitrogen and chlorophyll content in wheat (Triticum aestivum L.) Kragujevac J. Sci. 31 (2009) 69-74). Our recent study (Ref. 49, Yotsova et al., 2020) showed that increasing N content in the nutrient solution gradually reduces the roots and shoots growth of 14-day-old wheat seedlings of cv. Slomer. Another study by Tolley and Mohammadi (2020), doi:10.3390/plants9020144) also found variation in root and shoot dry weight of 14-day-old wheat seedlings, hydroponically grown on 4 mM and 0.5 mM nitrogen.

Reviewer 2 Report

Comments on Manuscript Plants: Manuscript ID: plants- 1098433

Full Title: Optimal Nitrogen Supply Ameliorates the Performance of Wheat Seedlings under Osmotic Stress in Genotype-Specific Manner

General comments

The subject is relevant. It is important to study responses of the different genetic background to environmental challenges. The conclusion was that better performance of the modern variety conceivably indicated that semi-dwarfing (Rht) alleles might have beneficial effect in arid regions and N deficiency conditions.

It is not clear what the basis of choosing the two cultivars was. Furthermore, it would be beneficial to know something about the yields of the cultivars, especially under the studied stress conditions.  

Detailed comments

Page 2, line 90: “Other authors reported that varying N application did not affect physiological functions under drought stress [ref 13].” They studied grain yield responses to drought of 30 wheat cultivars. Genotypes were prioritize for high yield under different water conditions. I can’t see N application.

Page 2, line 94: “… , in wheat, the introduction of della encoding mutant alleles…has affected the photosynthetic machinery and photosynthetic efficiency, as well as increased the plant nutrient efficiency [24].”

Page 3, line  99: “In the present study, we compared two wheat genotypes – an elite semi-dwarf, carrier of two Rht alleles (Enola), and an old tall one (Slomer), a selection from a landrace, with respect to their responses to N deficiency and drought, applied individually and in combination.“

The study of individual N deficiency is already published by the authors (ref 25) as already claimed in Page, line 105:

Page 3, line 105: “Our working hypothesis was that: 1) the representatives of the old germplasm and the advanced modern breeding might have different resistance mechanisms, and 2) that proper N level would support plant growth and would lead to better performance under water shortage. It was based on recent findings [25] on the impact of  N supply on physiological processes and some functional aspects of N nutrition and the photosynthesis in these two contrasting genotypes.”

Page 4, paragraph 2.1: Please be consistent by mentioning the cultivars: First the dwarf one (Enola) and following by the second old tall cultivar (Slomer).

Page 4, Figure 1: What is the difference between shoot length (Panel A) and seedling length (Panel B)?

Furthermore, the two cultivars responding differently to the N treatment compared to the published paper (ref 25). Actually Slomer root length just opposite in the published paper (control vs low N).

Please evaluate Fig 7 and 8 and the previously published Fig 5 (ref 25) on the conclusion of the photosynthesis effectivity.

Author Response

Q: It is not clear what the basis of choosing the two cultivars was. Furthermore, it would be beneficial to know something about the yields of the cultivars, especially under the studied stress conditions.  

A: We are thankful to Reviewer 2 for his precise remarks and have followed his recommendations accordingly. The corrections are given with Track changes.

In Introduction we have added a brief paragraph explaining the purpose for choosing the two wheat genotypes (page 3, line 103-131). In Materials and methods, the characteristics of the two wheat varieties are also mentioned. Parameters of the yield of these wheat varieties were discussed in two previous works (Ref. 35, 45).

Q: Detailed comments

Page 2, line 90: “Other authors reported that varying N application did not affect physiological functions under drought stress [ref 13].” They studied grain yield responses to drought of 30 wheat cultivars. Genotypes were prioritize for high yield under different water conditions. I can’t see N application.

A: The sentence is deleted.

Q: Page 2, line 94: “in wheat, the introduction of della encoding mutant alleles…has affected the photosynthetic machinery and photosynthetic efficiency, as well as increased the plant nutrient efficiency [24].”

A: We have substituted ‘plant nutrient (use) efficiency’ with ‘plant biomass under osmotic stress’ in the sentence.

Q: Page 3, line 99: “In the present study, we compared two wheat genotypes – an elite semi-dwarf, carrier of two Rht alleles (Enola), and an old tall one (Slomer), a selection from a landrace, with respect to their responses to N deficiency and drought, applied individually and in combination.“

A: We have outlined the differences between the two genotypes in the revised MS.

Q: The study of individual N deficiency is already published by the authors (ref 25) as already claimed in Page, line 105: “Our working hypothesis was that: 1) the representatives of the old germplasm and the advanced modern breeding might have different resistance mechanisms, and 2) that proper N level would support plant growth and would lead to better performance under water shortage. It was based on recent findings [25] on the impact of N supply on physiological processes and some functional aspects of N nutrition and the photosynthesis in these two contrasting genotypes.”

A: In our previous study (Ref. [25]) the focus was on nitrogen assimilation and plants were grown on nitrogen deficiency conditions (extremely low N supply (0.75 mM N or 0.1 of control N), while the low N level used in the present study was 1.5 mM N (0.2 of control N). In the revised manuscript we have pointed out these differences.

Q: Page 4, paragraph 2.1: Please be consistent by mentioning the cultivars: First the dwarf one (Enola) and following by the second old tall cultivar (Slomer).

A: In Figure 1C and D the mentioning of the two varieties was altered as the Reviewer appropriately suggested - first Enola and then Slomer, following the order they appear in the Figure.

Q: Page 4, Figure 1: What is the difference between shoot length (Panel A) and seedling length (Panel B)?

A: We thank the Reviewer for this remark. Panels A and B in Figure 1 must have the same inscription, namely "shoot length" and it was corrected in the revised MS.

Q: Furthermore, the two cultivars responding differently to the N treatment compared to the published paper (ref 25). Actually, Slomer root length just opposite in the published paper (control vs low N).

A: Regarding the responses of the two cultivars with respect to growth parameters observed in our previous paper and the present one, we could only speculate since low N levels used in the two studies were different: extremely low (N deprivation) 0.75 mM N in [25] and moderately low 1.5 mM N in the present work.  

It was shown that plant responses to low N or N deficiency are different and the effect of N deprivation or low N on root branching varies depending on the nitrogen nutrition situation of the plant itself and the degree to which the plants are stressed [Sun C-H, Yu J-Q and Hu D-G (2017) Nitrate: A crucial signal during lateral roots development. Front. Plant Sci. 8:485. doi: 10.3389/fpls.2017.00485].

Q: Please evaluate Fig 7 and 8 and the previously published Fig 5 (ref 25) on the conclusion of the photosynthesis effectivity.

A: As already mentioned above, low N supply was different in the two articles and effects could hardly be compared. Recent study by Shao et al. [2020] also showed different effects on the photochemistry of photosystem II (using OJIP test) of ryegrass seedlings at a moderately low N level (2.0 mM N) and ultra-low N level (0.5 mM N). Under control conditions, Chl fluorescence parameters of ryegrass leaves grown on 2.0 mM N were different from those grown on 0.5 mM and 5 mM N (Figures 2A and 3A in Shao et al. 2020). The authors observed that moderately low N could alleviate the inhibition of annual ryegrass growth by salt stress through a series of response mechanisms, whereas ultra-low N could not promote, but seriously inhibit the growth of annual ryegrass.

(Shao, A. et al., Moderately low nitrogen application mitigate the negative effects of salt stress on annual ryegrass seedlings. PeerJ  2020, 8, e10427. https://doi.org/10.7717/peerj.10427).

We have added the following sentence in discussion of the revised MS (page 12, line 355-359):

“Under control conditions, the quantum yield of regulated non-photochemical energy loss in PSII (ФNPQ) was higher in Enola than Slomer at normal N supply (Figure 8A, B) as observed previously [25]. However, under low N (0.2N) both cultivars showed different alterations than those observed under extremely low N (0.1N) [25]. “

Round 2

Reviewer 2 Report

Authors answered/explained all concerns raised and revised the manuscript accordingly